# Discovery and Characterization of a Novel Umbravirus from *Paederia scandens* Plants Showing Leaf Chlorosis and Yellowing Symptoms

**DOI:** 10.3390/v14081821

**Published:** 2022-08-19

**Authors:** Lianshun Zheng, Shuai Fu, Yi Xie, Yang Han, Xueping Zhou, Jianxiang Wu

**Affiliations:** 1State Key Laboratory of Rice Biology, Institute of Biotechnology, Zhejiang University, Hangzhou 310058, China; 2Hainan Institute, Zhejiang University, Sanya 572025, China; 3Beijing Institute of Genomics, Chinese Academy of Sciences, China National Center for Bioinformation, Beijing 100101, China; 4State Key Laboratory for Biology of Plant Diseases and Insect Pests, Institute of Plant Protection, Chinese Academy of Agricultural Sciences, Beijing 100193, China

**Keywords:** novel virus, umbravirus, viral infectious clone, RNA silencing suppressor, *Paederia scandens*

## Abstract

Umbraviruses are a special class of plant viruses that do not encode any viral structural proteins. Here, a novel umbravirus that has been tentatively named *Paederia scandens* chlorosis yellow virus (PSCYV) was discovered through RNA-seq in *Paederia scandens* plants showing leaf chlorosis and yellowing symptoms. The PSCYV genome is a 4301 nt positive-sense, single strand RNA that contains four open reading frames (ORFs), i.e., ORF1–4, that encode P1–P4 proteins, respectively. Together, ORF1 and ORF2 are predicted to encode an additional protein, RdRp, through a −1 frameshift mechanism. The P3 protein encoded by ORF3 was predicted to be the viral long-distance movement protein. P4 was determined to function as the viral cell-to-cell movement protein (MP) and transcriptional gene silencing (TGS) suppressor. Both P1 and RdRp function as weak post-transcriptional gene silencing (PTGS) suppressors of PSCYV. The PVX-expression system indicated that all viral proteins may be symptom determinants of PSCYV. Phylogenetic analysis indicated that PSCYV is evolutionarily related to members of the genus *Umbravirus* in the family *Tombusviridae*. Furthermore, a cDNA infectious clone of PSCYV was successfully constructed and used to prove that PSCYV can infect both *Paederia scandens* and *Nicotiana benthamiana* plants through mechanical inoculation, causing leaf chlorosis and yellowing symptoms. These findings have broadened our understanding of umbraviruses and their host range.

## 1. Introduction

Umbraviruses are positive-sense, single strand RNA viruses belonging to the genus *Umbravirus* in the family *Tombusviridae* [1]. Umbraviruses differ from most other viruses in that the genomes of umbraviruses lack the open reading frames (ORFs) encoding viral coat protein (CP), which makes them unable to form conventional virus particles by themselves [1]. Under natural circumstances, the survival and transmission of umbraviruses is usually dependent on the assistance of helper viruses, which are usually members from the family *Luteoviridae* [1,2,3,4]. Umbraviruses can spread through mechanical inoculation and replicate in infected plants, but they can also be encapsulated by the CPs of luteoviruses, which allows the umbravirus to be transmitted by aphids in the field [1,4]. Some umbraviruses, such as groundnut rosette virus (GRV), have satellite RNAs, which are important for symptom development and/or aphid transmission [5,6,7].

Cap structures do not exist at the 5′ terminus of umbraviruses, and neither does the polyadenosine tail at the 3′ terminus. The umbravirus genome encodes four potential non-structural proteins [4]. ORF1, located at the 5′ end, encodes a putative protein [4]. ORF2, the second ORF following ORF1, encodes a protein containing eight conserved motifs of a viral RNA-dependent RNA polymerase (RdRp) [1,8]. Together, umbravirus ORF1 and ORF2 may encode a larger, putative RdRp through a mechanism of 1 frameshift [4]. ORF3 and ORF4 almost completely overlap. The ORF3 encoding protein can bind viral RNA to form filamentous ribonucleoprotein (RNP) complexes that protect the viral genome from RNA degradation, and assists in the systemic infection of the virus [1,9,10,11,12]. The ORF4 encoding protein has the characteristics of the cell-to-cell movement protein (MP) of plant viruses. For example, the ORF4 encoding proteins of GRV and carrot mottle virus (CMoV) can localize to plasmodesmata (PD) and induce the formation of tubular structures on protoplasts’ surface [13,14,15]. GRV ORF4 encoding protein can partly replace the MP function of potato virus X (PVX) or cucumber mosaic virus (CMV) [13,16]. SUMOylation, a kind of posttranslational modification, plays an important role in the targeting of umbravirus ORF4 encoding protein to PDs [15].

Many traditional detection methods are widely used in the detection of plant RNA viruses, especially serological detection represented by enzyme-linked immunosorbent assay (ELISA) and molecular detection represented by RT-PCR [17]. However, these techniques fail to detect unknown viruses in the field. With the development of high-throughput RNA sequencing (RNA-seq) technique and bioinformatics, many novel viruses infecting different organisms have been identified [18,19]. For example, using the RNA-seq technique, our laboratory has identified several novel viruses in field-collected plants such as rice curl dwarf-associated picornavirus [20], rice-associated noda-like virus 1 and 2 [21], rice dwarf-associated bunya-like virus [22], and wheat yellow stunt-associated betaflexivirus [23].

*Paederia scandens* (Lour.) Merr., a perennial herbaceous vine belonging to the genus *Paederia* in the family *Rubiaceae*, is widely distributed in China, Vietnam, India, Japan, and the USA [24,25]. This plant is used in traditional Chinese medicine for the treatment of multiple diseases [24]. Compounds from *Paederia scandens* such as iridoid glucosides, flavonoids, and volatile oils possess many pharmacological effects, such as anti-nociceptive, anti-inflammatory, and anti-tumor [24,26]. So far, there have not been any studies that have reported virus infection in *Paederia scandens* plants. 

In 2021, during plant viral disease surveys in the China National Botanical Garden in Beijing City, China, some *Paederia scandens* plants were discovered with virus-like symptoms of leaf chlorosis and yellowing. In this study, we identified and characterize a novel umbravirus from symptomatic *Paederia scandens* plants.

## 2. Materials and Methods

### 2.1. Plant Materials

In the summer of 2021, *Paederia scandens* plants showing leaf chlorosis and yellowing symptoms were collected from the China National Botanical Garden in Beijing city, China. The 16c transgenic *N. benthamiana* seeds used in this study were kindly provided by David C. Baulcombe (University of Cambridge, Cambridge, UK) [27]. The RFP-H2B transgenic *N. benthamiana* seeds were kindly provided by Michael M. Goodin (University of Kentucky, Lexington, USA) [28]. The 16-TGS plants used were described previously [29]. The *Paederia scandens* seedlings used in this work were purchased from Gang Meizi Trade Co., Ltd. (Yangjiang, China). Plants were grown in a growth chamber maintained at 25 °C (day) and 18 °C (night), 60% relative humidity, and 16 h:8 h (light:dark) photoperiod.

### 2.2. RNA Sequencing (RNA-seq) and De Novo Assembly

The RNA-seq and de novo assembly was performed as previously described [21,22,23]. Briefly, total RNA was extracted from the sampled leaf tissues using the EASYspin Plus Complex Plant RNA Kit (Aidlab Biotech, Beijing, China). Ribosomal RNAs were removed by the TruSeq RNA Sample Prep Kit (Illumina, San Diego, CA, USA). An RNA library was constructed using the TruSeq RNA Sample Prep Kit according to the manufacturer’s protocol (Illumina, San Diego, CA, USA), followed by sequencing on the Illumina HiSeq X-ten platform (Biomarker Technologies, Beijing, China). Low-quality reads were filtered and adapters of the paired-end raw reads were trimmed using the CLC Genomics Workbench 9.5 software (QIAGEN, Hilden, Germany). The clean reads were de novo assembled into contigs using the Trinity v2.3.2 program (Broad Institute and the Hebrew University of Jerusalem, Cambridge, Massachusetts and Jerusalem, USA and Israel) [22].

### 2.3. RT-PCR and Rapid Amplification of cDNA Ends (RACE)

Total RNA was extracted from the plant samples utilizing the RNAiso Plus reagent following the manufacturer’s protocol (TaKaRa, Tokyo, Japan). Reverse transcription (RT) was performed with M-Mlv (TaKaRa, Tokyo, Japan) and specific reverse primers, followed by PCR using Phanta DNA polymerase (Vazyme, Nanjing, China) and specific primer pairs (Appendix A). To amplify the 3′ and 5′ end sequences of the newly identified virus, 5′/3′-RACE was carried out utilizing the HiScript-TS 5′/3′ RACE Kit (Vazyme) according to the manufacturer’s instructions. The resulting PCR products were gel purified and Sanger sequenced. Based on the assembled viral genomic sequence, the full-length genome of the newly identified virus was obtained from the diseased *Paederia scandens* plants through RT-PCR, cloned, and then Sanger sequenced. For plasmid construction, transformants were identified by PCR with the KOD One PCR Master Mix (TOYOBO, Osaka, Japan) and specific primers for viral genomic or ORF sequences. 

### 2.4. Analyses of Viral Genome Organization and Proteins 

The ORFs in the viral genomic RNA of the novel virus were predicted using the ORFfinder service (http://www.ncbi.nlm.nih.gov/orffinder, accessed on 10 December 2021) and utilizing the carrot mottle virus (CMoV) genome (NC_011515.1) as a model. Amino acid (aa) sequences of the predicted proteins encoded by the ORFs were subjected to BLASTp searches in the GenBank database. Domains in the predicted proteins were identified using the Conserved Domain Search Service (CD-Search) software in the NCBI (https://www.ncbi.nlm.nih.gov/Structure/cdd/wrpsb.cgi, accessed on 10 January 2022). Sequence identities of the novel virus genomic RNA and its proteins were aligned with that of known umbraviruses downloaded from the GenBank database using the DNASTAR7.1 software (https://www.dnastar.com/, accessed on 22 February 2022). Detailed information on these sequences downloaded from the GenBank database is provided in Appendix A.

### 2.5. Phylogenetic Analysis

The RdRp sequences of representative viruses from different genera of the family *Tombusviridae* and two other viruses from the family *Solemoviridae* were downloaded from the GenBank database, and their detailed information is presented in Appendix A. RdRp aa sequences of the novel virus and these downloaded viruses were aligned utilizing MUSCLE in the MEGA X software (https://www.megasoftware.net/, accessed on 15 February 2022). The phylogenetic analysis was performed using the maximum-likelihood method in the MEGA X package with 1000 bootstrap replicates (https://www.megasoftware.net/ (accessed on 15 February 2022)).

### 2.6. Plasmid Construction

The predicted viral genes were amplified by RT-PCR from total RNA of the infected *Paederia scandens* plants using the specific primers listed in Appendix A. The amplified genes were separately subcloned into the expression vector pGD, pCambia 1300-Flag, pCambia 1300-GFP, pGR106, or pCB301-RZ using the One-Step Cloning kit per the manufacturer’s protocol (Vazyme). The constructed vectors were confirmed by Sanger sequencing.

### 2.7. Agroinfiltration Assays

The constructed expression vectors were separately transformed into *Agrobacterium tumefaciens* strain EHA105 or GV3101 via electroporation. Before infiltration, the cultured *A. tumefaciens* was diluted with the infiltration buffer (10 mM MgCl2, 10 mM MES, pH 5.8, and 100 μM acetosyringone) up to OD_600_ = 0.8. For co-infiltration assays, two diluted *A. tumefaciens* harboring two different constructs were mixed at the same volume ratio and then infiltrated into the leaves of *N. benthamiana* plants with a needle-less syringe.

### 2.8. Viral Movement Protein Complementation Assay

The PVX-GFPΔp25 construct expressing a movement-defective potato virus X was obtained from a previous study [30]. For the positive control, the cultured *A. tumefaciens* strain GV3101 carrying the PVX-GFPΔp25 construct was prepared and diluted till OD_600_ = 0.00125 with the infiltration buffer, and then mixed 1:1:1 (*v/v/v*) with *A. tumefaciens* carrying a construct expressing the rice stripe virus (RSV)-encoded NSvc4 protein (OD_600_ = 0.8) and *A. tumefaciens* carrying a construct expressing the tomato bushy stunt virus (TBSV) P19 protein (OD_600_ = 0.8). For other treatments, the cultured *A. tumefaciens* carrying the construct expressing RSV NSvc4 was replaced by the cultured *A. tumefaciens* carrying the construct expressing β-glucuronidase (GUS) or the *A. tumefaciens* carrying the construct expressing P4 protein of the new virus. GFP fluorescence in co-infiltrated *N. benthamiana* leaves was observed and photographed under a confocal microscope (model FLUOVIEW FV3000; Olympus, Tokyo, Japan) and a UV lamp (UV Products, Upland, CA, USA).

### 2.9. Western Blot Assays

Total protein was extracted from the leaf tissues using an extraction buffer (50 mM Tris-HCl, pH 6.8, 9 M carbamide, 4.5% SDS, and 7.5% 2-mercaptoethanol). The resulting sample proteins were separated by electrophoresis in 12.5% sodium dodecyl sulfate-polyacrylamide gel. After transferring proteins onto nitrocellulose membranes, Western blot assays were performed as previously described [31].

### 2.10. Mechanical Inoculation

To prepare the tissue homogenate for inoculation, diseased leaves from the field-collected PSCYV-infected *Paederia scandens* plants or the agro-infiltrated *N. benthamiana* plants with typical symptoms were ground to slurry in 0.1 M pH 7.2 phosphate buffer using emery powder. For mechanical inoculation, leaves of plants were sprinkled with an appropriate amount of emery powder, and then gently rubbed using an index finger that had been immersed in the tissue homogenate.

### 2.11. Confocal Microscope

The agro-infiltrated leaves from the *N. benthamiana* plants were observed under a confocal microscope (Olympus). The excitation wavelengths of RFP, GFP, and chloroplast autofluorescence were separately set at 561, 488, and 640 nm, and the emission wavelength was set at 583 nm for RFP, 510 nm for GFP, and 671 nm for chloroplast.

## 3. Results

### 3.1. Identification of a Novel Umbravirus in Paederia Scandens

In the summer of 2021, during a survey of plant viral diseases in the China National Botanical Garden in Beijing City, China, some *Paederia scandens* plants showing leaf chlorosis and yellowing symptoms (Figure 1A) were found and collected for virus identification through RNA-seq. Analyses of the resulting sequence reads using the TopHat software v2.1.1 produced a long contig of 4293 nucleotides. A BLASTx search of the GenBank database with this long contig query found that the predicted protein from this long contig shared the highest aa sequence identity of 71.27% (60% sequence coverage) with the RdRp from Ethiopian tobacco bushy top virus (ETBTV). The above results indicated the presence of an unknown virus in the tested *Paederia scandens* plants, which could be a member of the genus *Umbravirus*. To confirm this RNA-seq result, a partial segment of this long contig was amplified from total RNA extracted from the collected plant using RT-PCR, and an expected gene segment of 535 bp was amplified (Figure 1B). Then, 5′ and 3′ RACEs were performed to obtain the full-length genome sequence of this novel virus. The results of the 5′ and 3′ RACEs and Sanger sequencing showed that the genomic RNA of the novel virus contained 4301 nt (Figure 2). The complete genome sequence of this novel virus has been deposited in the GenBank under the accession number OP053684. The genome sequence of this novel virus shared a 47.9–57.4% nt sequence identity with that of ten more homologous umbraviruses, with the most homologous virus as tobacco mottle virus (TMoV) (Table 1). This genomic sequence difference exceeded the cutoff value for species discrimination in the genus *Umbravirus* (i.e., 70% for the nt sequence identity) [4]. Thus, this newly identified virus is believed to be a new species in the genus *Umbravirus*. Based on its genome and typical viral symptoms, this novel virus has been tentatively named *Paederia scandens* chlorisis yellow virus (PSCYV).

### 3.2. Genome Organization and Protein Prediction of PSCYV

To determine viral genome organization and proteins, the ORFs in the PSCYV genomic RNA and viral proteins of PSCYV were predicted. The results showed that the genomic RNA of PSCYV contained a 30-nt 5′-untranslated region (UTR), four ORFs, and a 717-nt 3′-UTR without a poly(A) tail (Figure 2). ORF1 of PSCYV (nt position 31–999) was predicted to encode a putative protein of 34.6 kDa (Figure 2), sharing the highest aa sequence identity (35.8%) with the P1 of opium poppy mosaic virus (OPMV) (Table 1). ORF2 (nt position 1539–2660) was predicted to encode a protein that contained a conserved domain: RdRP_3 (nt position 1071–2465, pfam00998) (Figure 2), and shared the highest aa sequence identity (71.6%) with the P2 of ETBTV (Table 1). An 876 aa protein of 97.7 kDa was encoded by ORF1 together with ORF2 through the -1 frameshift mechanism (Figure 2), and shared the highest aa sequence identity (60.6%) with the RdRp of tobacco mottle virus (TMoV) (Table 1). Therefore, this protein was considered to be the RdRp of PSCYV. ORF3 (nt position 2792–3499) was predicted to encode a long-distance movement protein of 25.6 kDa (Figure 2) that contained a Umbravirus_LDM super family domain (nt position 2792–3478, cl04774), and shared the highest aa sequence identity (40.0%) with that of groundnut rosette virus (GRV) (Table 1). ORF4 (nt position 2808–3584) was predicted to encode a cell-to-cell movement protein (MP) of approximately 28.6 kDa that contained a 3A super family domain (nt position 2823–3371, cl02970) (Figure 2), and shared the highest aa sequence identity (58.4%) with the MP of tobacco bushy top virus (TBTV) (Table 1).

### 3.3. Phylogenetic Analysis of PSCYV

To illustrate the evolutionary relationships of PSCYV with other viruses, a phylogenetic tree based on the RdRp aa sequences of PSCYV, 38 representative viruses in different genera of the family *Tombusviridae,* and 2 other viruses of the family *Solemoviridae* from the NCBI database (Appendix A) was constructed. The resulting phylogenetic tree indicated that PSCYV was clustered in a clade together with viruses from the genus *Umbravirus*, and evolutionarily closest to TBTV (Figure 3).

### 3.4. Subcellular Localization of Viral Proteins and Identification of Viral Movement Protein

To analyze subcellular localization patterns of viral proteins, PSCYV *P1*, *P2*, *RdRp*, *P3,* and *P4* genes were respectively subcloned into the transient expression vector pCambia 1300-GFP to produce recombinant expression vectors, pCambia 1300-P1-GFP (P1:GFP), pCambia 1300-P2-GFP (P2:GFP), pCambia 1300-RdRp-GFP (RdRp:GFP), pCambia 1300-P3-GFP (P3:GFP), and pCambia 1300-P4-GFP (P4:GFP). *N. benthamiana* leaves were separately agro-infiltrated with these constructed expression vectors, and the green fluorescence from the fusion protein was observed under a confocal microscope at 2 d post infiltration (dpi). The results showed that all P1, P2, and RdRp were located in the cytoplasm and had completely co-localized with chloroplast (red signal, chloroplast autofluorescence) (Figure 4A). P3 formed puncta in cytoplasm and nucleus (Figure 4B).

Plant viruses can change the structure of plasmodesmata (PD) in different ways, and enlarge the aperture of PD to obtain cell to cell movement in plants [32]. To investigate whether PSCYV P4 can target PD in the cell walls, P4-GFP and TMV MP-RFP fusion proteins were co-expressed in *N. benthamiana* leaves through agroinfiltration. Observation with confocal microscopy found that P4 was located in the cytomembrane, cytoplasm, and nucleus, in addition to forming puncta on the cytomembrane (Figure 4C). Furthermore, P4 co-localized with TMV-MP (red punctate signal) (Figure 4C). Then, 1 M NaCl was injected into the leaves of co-infiltrated *N. benthamiana*, followed by immediate observation of the fluorescence under a confocal microscope. P4 and TMV MP were found to co-localize at PDs between cells (Figure 4C), which indicated that PSCYV P4 can target PDs. 

To further prove whether P4 is the viral cell-to-cell movement protein of PSCYV, *P4* was subcloned into the expression vector pCambia 1300-Flag to express the P4-flag fusion protein. Transformed *A. tumefaciens* culture harboring the GUS-flag (the negative control), P4-flag, or RSV NSvc4 (the positive control) vector was mixed with *A. tumefaciens* culture harboring the TBSV P19 expression vector and *A. tumefaciens* culture harboring the PVX-GFPΔp25 expression vector. The mixed *A. tumefaciens* cultures were separately infiltrated into one side of *N. benthamiana* leaves, and the infiltrated leaves were observed under a confocal microscope and a UV lamp at 7 dpi. We found that PSCYV P4, like RSV NSvc4, could help the mutant PVX-GFPΔp25 to move into adjacent cells, but GUS-flag could not (Figure 5A,B). This result indicates that P4 can function as a viral movement protein (MP). Western blot assay analysis using a GFP-specific antibody demonstrated that more GFP accumulation was detected in the *N. benthamiana* leaves expressing P4-flag or RSV NSvc4, compared to the GUS-flag control (Figure 5C), which further confirmed that PSCYV P4 can function as a cell-to-cell movement protein. The expression of P4-flag and GUS-flag was confirmed by Western blot assays using an anti-flag antibody, and NSvc4 expression was confirmed by Western blot assay using a NSvc4-specific antibody.

### 3.5. Pathogenicity of Viral Proteins of PSCYV

The potato virus X (PVX)-based vector is usually used to express target genes in PVX host plants [33,34]. To determine the symptom determinant of PSCYV, five PVX-based recombinant expression vectors were constructed. The PVX-P1, PVX-P2, PVX-P3, PVX-P4, and PVX-RdRp vectors respectively contained the PSCYV *P1*, *P2*, *P3*, *P4*, or *RdRp* gene. These resulting recombinant expression vectors were separately agro-infiltrated into *N. benthamiana* leaves. *A. tumefaciens* harboring empty PVX vector was used as the negative control. At 10 dpi, *N. benthamiana* leaves infiltrated with PVX-P4 or PVX-RdRp developed severely necrotic symptoms, while *N. benthamiana* leaves infiltrated with PVX-P1, PVX-P2, or PVX-P3 showed chlorosis, mottle, and slight punctate necrosis, which are somewhat different from the mosaic symptom caused by PVX infection (Figure 6A). At 21 dpi, *N. benthamiana* leaves infiltrated with PVX-P4 showed more severely necrotic symptoms (Figure 6A). However, PVX-RdRp-infiltrated plants showed symptom recovery, which was the same as the empty PVX vector infiltrated plants at 21 dpi (Figure 6A). PVX-P1-, PVX-P2-, or PVX-P3-infiltrated plants still showed chlorosis and mottle symptoms at 21 dpi (Figure 6A), and the punctate necroses became more obvious at the leaves of PVX-P1-infiltrated plants (Figure 6A). 

RT-PCR was used to detect the presence of viral genes from the upper leaves of infiltrated plants at 21 dpi. The RT-PCR result showed that the *P1, P2, P3,* and *P4* genes could be amplified, but not the *RdRp* gene (Appendix A). This indicated that the PVX-RdRp ceased to exist at 21 dpi. The reason for the symptom recovery of PVX-RdRp-infected plants in the late infection stage was thought to be that PSCYV RdRp is too large (97.7 kDa) and is discarded by PVX.

To determine the effect of viral proteins on the accumulation of PVX in infected plants, Western blot assay was performed using PVX CP-specific antibody. The results demonstrated that the PVX accumulation levels in PVX-P4- or PVX-P1-infected plants were significantly higher than that in PVX-infected plants at 21 dpi (Figure 6B). However, PVX accumulation levels in PVX-P1-infected plants were significantly lower than that in PVX-infected plants at 10 dpi (Figure 6B). PVX accumulation levels in PVX-P2 and PVX-P3-infected plants were slightly higher than that PVX-infected plants at 21 dpi (Figure 6B). PVX accumulation level in the PVX-RdRp-infected plants was the same as the PVX-infected plants at 10 dpi, but was significantly lower than that in the PVX-infected plants at 21 dpi (Figure 6B). These above results suggest that P1, P2, P3, P4, and RdRp all play an important role in the symptom development of infected plants and may be symptom determinants of PSCYV.

### 3.6. The TGS and PTGS Suppressors of PSCYV

Plants can use the transcriptional gene silencing (TGS) and post-transcriptional gene silencing (PTGS) to resist virus infections, whereas plant viruses encode TGS and PTGS suppressors to overcome this plant innate defense [34,35,36,37,38]. To determine the TGS suppressor of PSCYV, we infiltrated 16-TGS *N. benthamiana* plants with *A. tumefaciens* cultures carrying the PVX-P1, PVX-P2, PVX-P3, PVX-P4, or PVX-RdRp expression vector. The PVX-βC1 and PVX vector were used as the positive and negative control, respectively. At 12 dpi, systemic leaves of 16-TGS *N. benthamiana* plants individually ago-infiltrated with PVX-P1, PVX-P2, PVX-P3, or PVX-RdRp remained red under UV light, which was the same as the PVX negative control (Figure 7A). However, the systemic leaves of 16-TGS *N. benthamiana* plants agro-infiltrated with PVX-P4 or PVX-βC1 showed obvious green fluorescence (Figure 7A). Western blot assay analysis showed that accumulation level of GFP in the leaves agro-infiltrated with PVX-4 or PVX-βC1 was significantly higher than that in the leaves agro-infiltrated with PVX, PVX-P1, PVX-P2, PVX-P3, or PVX-RdRp (Figure 7B). These results indicate that PSCYV P4 is a viral suppressor of TGS.

To determine whether PSCYV encodes a PTGS suppressor, the leaves of wild-type *N. benthamiana* plants were infiltrated with a mixed *A. tumefaciens* culture carrying 35S-GFP and 35S-dsGFP, and P1, P2, P3, P4, RdRp, P19, or the empty pGD vector. The P19 and empty vector were separately used as the positive and negative control. At 4 dpi, the leaves of *N. benthamiana* infiltrated with the P1, P2, P3, P4, or RdRp lost their GFP fluorescence, but the leaves infiltrated with P19 still showed GFP fluorescence (Figure 7C). Western blot assay analysis using a GFP-specific antibody also revealed a negligible GFP accumulation in the leaves of *N. benthamiana* infiltrated with the P1, P2, P3, P4, RdRp, or the empty vector (Figure 7D). However, this assay also showed the high GFP accumulation in the leaves of *N. benthamiana* infiltrated with the positive control P19 (Figure 7D). These results indicate that PSCYV does not encode a silencing suppressor that can suppress RNA silencing induced by dsRNA.

To further identify the PTGS suppressor of PSCYV, we constructed five transient expression vectors using the pCambia 1300-Flag vector that respectively expresses the P1-flag, P2-flag, P3-flag, P4-flag, and RdRp-flag fusion protein. The GFP-transgenic *N. benthamiana* 16c plants were infiltrated with a mixed *A. tumefaciens* culture carrying ssGFP and P1-flag, P2-flag, P3-flag, P4-flag, RdRp-flag, p19, or GUS-flag construct at a 1:1 ratio. The P19 and GUS-flag were separately used as the positive and negative control. The mixed *A. tumefaciens* was separately infiltrated into one side of 16c *N. benthamiana* leaves, and the infiltrated leaves were observed under a UV lamp at 5 dpi. The results showed that the leaves infiltrated with the P19 remained strong green GFP fluorescence and the leaves infiltrated with GUS-flag lost green GFP fluorescence (Figure 7E). The leaves infiltrated with P1-flag or RdRp-flag showed weak green GFP fluorescence, while the leaves infiltrated with P2-flag, P3-flag, or P4-flag lost the GFP fluorescence, which was similar to the negative control (Figure 7E). Western blot assay analyses using a GFP-specific antibody also revealed a negligible amount of GFP accumulation in the leaves of 16c *N. benthamiana* infiltrated with the P2-flag, P3-flag, P4-flag, or GUS-flag, but a higher GFP accumulation in the leaves infiltrated with the P19, P1-flag, or RdRp-flag. The effect of P1-flag and RdRp-flag appeared to be weaker than that of P19 (Figure 7F). These results indicated that both P1 and RdRp function as weak silencing suppressors that can suppress RNA silencing induced by ssRNA.

### 3.7. Infectious Clone Construction of PSCYV and Mechanical Inoculation 

The leaves from the field-collected PSCYV-infected *Paederia scandens* plant were homogenized and used to mechanically inoculate the leaves of *N. benthamiana* plants. The chlorosis and yellowing symptoms seen on the collected PSCYV-infected *Paederia scandens* plants also appeared on the leaves of the inoculated *N. benthamiana* plants at 18 dpi (Figure 8A). Furthermore, the expected PCR segment of 535 bp was amplified from these plants by RT-PCR using PSCYV-specific primer pair (Figure 8B). These results demonstrate that PSCYV can infect *N. benthamiana* plants by mechanical inoculation.

The infectious clone is an important tool to study viral biological characteristics and genome function. To obtain an infectious clone of PSCYV, the full-length cDNA sequence of PSCYV genome was subcloned into the vector pCB301-RZ to produce a recombinant vector, pCB301-PSCYV. The leaves of *N. benthamiana* plants were infiltrated with *A. tumefaciens* culture harboring pCB301-PSCYV. At 18 dpi, the infiltrated *N. benthamiana* plants showed leaf chlorosis and yellowing symptoms (Figure 8A) that are similar to that of *N. benthamiana* plants that were mechanically inoculated with the homogenate of PSCYV-infected *Paederia scandens* plants. The systemic leaves of the infiltrated *N. benthamiana* plants were harvested and the presence of PSCYCV was detected by RT-PCR using the PSCYV-specific primer pair, and the expected gene fragment of 535 bp was also amplified from the infiltrated *N. benthamiana* plants (Figure 8B). This result indicates that the leaf chlorosis and yellowing symptoms of inoculated *N. benthamiana* plants were caused by PSCYV infection, and the constructed infectious clone of PSCYV was viable. To prove Koch’s postulates and determine whether PSCYV can infect the *Paederia scandens* host, healthy *Paederia scandens* plants were mechanically inoculated with the homogenate from the agro-infiltrated *N. benthamiana* plants. At 18 dpi, the inoculated *Paederia scandens* plants also showed chlorosis and yellowing symptoms on leaves (Figure 8C). Furthermore, PSCYV infection in these inoculated plants was confirmed by RT-PCR detection (Figure 8D). These results clearly revealed that the cDNA infectious clone of PSCYV was successfully created and PSCYV can infect *Paederia scandens* plants through mechanical inoculation and cause leaf chlorosis and yellowing symptoms.

## 4. Discussion

In this study, a novel virus in *Paederia scandens* plants showing leaf chlorosis and yellowing symptoms was discovered through RNA-seq, and it was tentatively named PSCYV. To date, PSCYV is the first reported virus that can infect *Paederia scandens* plants. The PSCYV genome is a positive-sense, single strand RNA that is 4301 nt in length. The genome organization of PSCYV is similar to that of previously reported umbraviruses. It contains four ORFs and encodes five speculated proteins: P1, P2, P3, P4, and RdRp. Together, ORF1 and ORF2 can co-encode an RdRp through a -1-frameshift mechanism. PSCYV is evolutionarily close to other viruses from the genus *Umbravirus* in the family *Tombusviridae.* The PVX heterologous expression system implied that all P1, P2, P3, P4, and RdRp may be symptom determinants of PSCYV (Figure 6). Furthermore, a cDNA infectious clone of PSCYV was successfully constructed, and PSCYV was found to infect *Paederia scandens* and *N. benthamiana* plants through mechanical inoculation, causing leaf chlorosis and yellowing symptoms. Thus, Koch’s postulate for PSCYV was proved.

In this work, P1, P2, and RdRp of PSCYV were found to co-locate at chloroplast through subcellular localization analyses. Ahquist et al. found that the plus strand RNA viruses can induce spherule formation on the membrane of host subcellular organelles, and replicate there [39]. Thus, we speculate that chloroplast membrane may be the replication site of PSCYV. 

Previous studies have shown that helper component-protease (HC-Pro) encoded by potyviruses and 2b protein of CMV function as the suppressors of TGS and PTGS, and promote virus systemic spread [36,37,38,40,41,42]. In this study, it was found that both P1 and RdRP proteins function as the weak PTGS suppressors of PSCYV, and P4 function as the TGS suppressor of PSCYV. This is the first study that reported the PTGS and TGS suppressors of umbraviruses. P3 of GRV, pea enation mosaic virus 2 (PEMV-2), and TMoV from umbraviruses can replace the role of TMV CP in viral long-distance movement [43]. Combined with domain analysis, PSCYV P3 was speculated to function as the viral long-distance movement protein.

In this study, PSCYV P4 was found to target PDs and act as a viral cell-to-cell movement protein. This finding is consistent with the function of P4 encoded by other two umbraviruses, GRV and CMoV [13,14,15,16]. CMoV P4 can interact with SCE1, SUMO1, and SUMO2 in the nucleus [15], which implies that the umbravirus P4 can enter nuclei and perform relevant functions. TGS is an important innate immune pathway, which usually occurs in the nucleus. In this present study, PSCYV P4 was also found to localize in nuclei and acts as a viral TGS suppressor (Figure 4C and Figure 7). Thus, it was hypothesized that umbravirus P4 can regulate the transcription of host genes in nuclei. However, this hypothesis remains to be further clarified.

Under the nature environment, genomic RNAs of umbraviruses are packaged with CPs encoded by members of the family *Luteoviridae*, and umbraviruses often co-infect plants with luteoviruses by aphid transmission [1,4]. Unfortunately, no member from the family *Luteoviridae* was discovered in the PSCYV-infected *Paederia scandens* plants by RNA-seq (data not shown). Therefore, whether there is a virus from the family *Luteoviridae* that co-infects *Paederia scandens* with PSCYV remains unknown, and the transmission of PSCYV in the field needs to be further studied.

In summary, a novel virus, *Paederia scandens* yellow mottle virus (PSCYV) belonging to the genus *Umbravirus,* the family *Tombusviridae*, has been identified in this work. PSCYV P4 was identified as a cell-to-cell movement protein and as TGS suppressor. Both P1 and RdRp were identified as weak PTGS suppressors of PSCYV. In addition, a cDNA infectious clone of PSCYV was successfully constructed, and PSCYV can infect *Paederia scandens* and *N. benthamiana* plants, causing leaf chlorosis and yellowing symptoms. These findings have broadened our understanding of umbraviruses and viruses infecting *Paederia scandens*, and can also provide an important theoretical basis for controlling this plant viral disease.

## Figures and Tables

**Figure 1 viruses-14-01821-f001:**
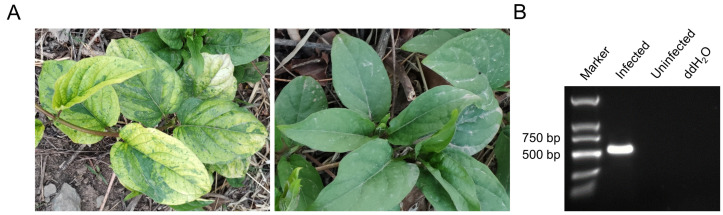
Symptoms and RT-PCR detection of *Paederia scandens* chlorosis yellow virus (PSCYV) infection. (**A**) Symptoms of PSCYV infection. The diseased *Paederia scandens* plant showed leaf chlorosis and yellowing symptoms (**left**). The image on the right shows a non-infected plant. (**B**) RT-PCR detection results of PSCYV infection in *Paederia scandens* plants using the specific primer pair designed according to the assembled long contig.

**Figure 2 viruses-14-01821-f002:**
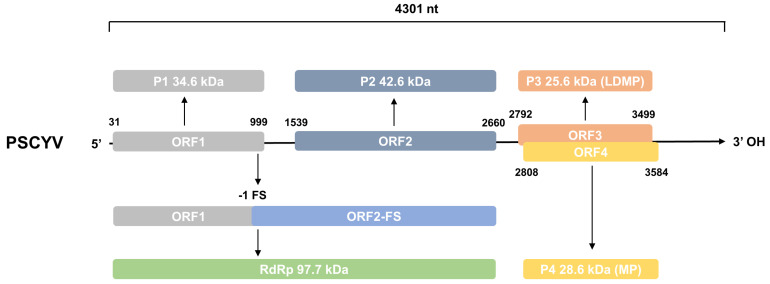
Schematic diagram of the PSCYV genome organization. Nucleotide position of individual protein is indicated above or under each gene box. The conserved umbraviral motifs are indicated above or under the diagram. ORF2-FS stands for ORF2 that is translated through -1 frameshift; -1 FS, -1 frameshift; RdRp, RNA-dependent RNA polymerase; LDMP, viral long-distance movement protein; and MP, viral cell-to-cell movement protein.

**Figure 3 viruses-14-01821-f003:**
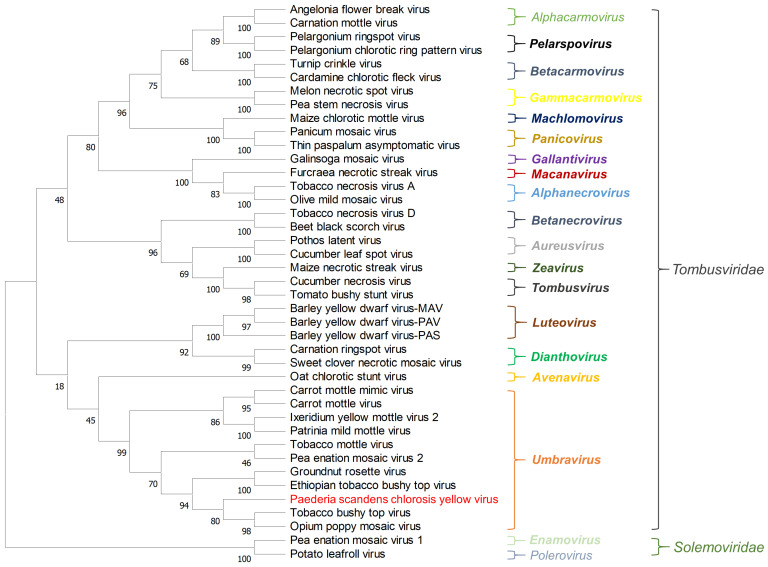
Phylogenetic relationships between PSCYV and other 40 related viruses. The phylogenetic tree was constructed using the aa sequences of RdRps and the maximum likelihood method with 1000 bootstraps. Viruses from the same genus are shown in the same color. PSCYV is shown in red. The bootstrap values are indicated adjacent to the nodes. Accession numbers of these sequences are listed in Appendix A.

**Figure 4 viruses-14-01821-f004:**
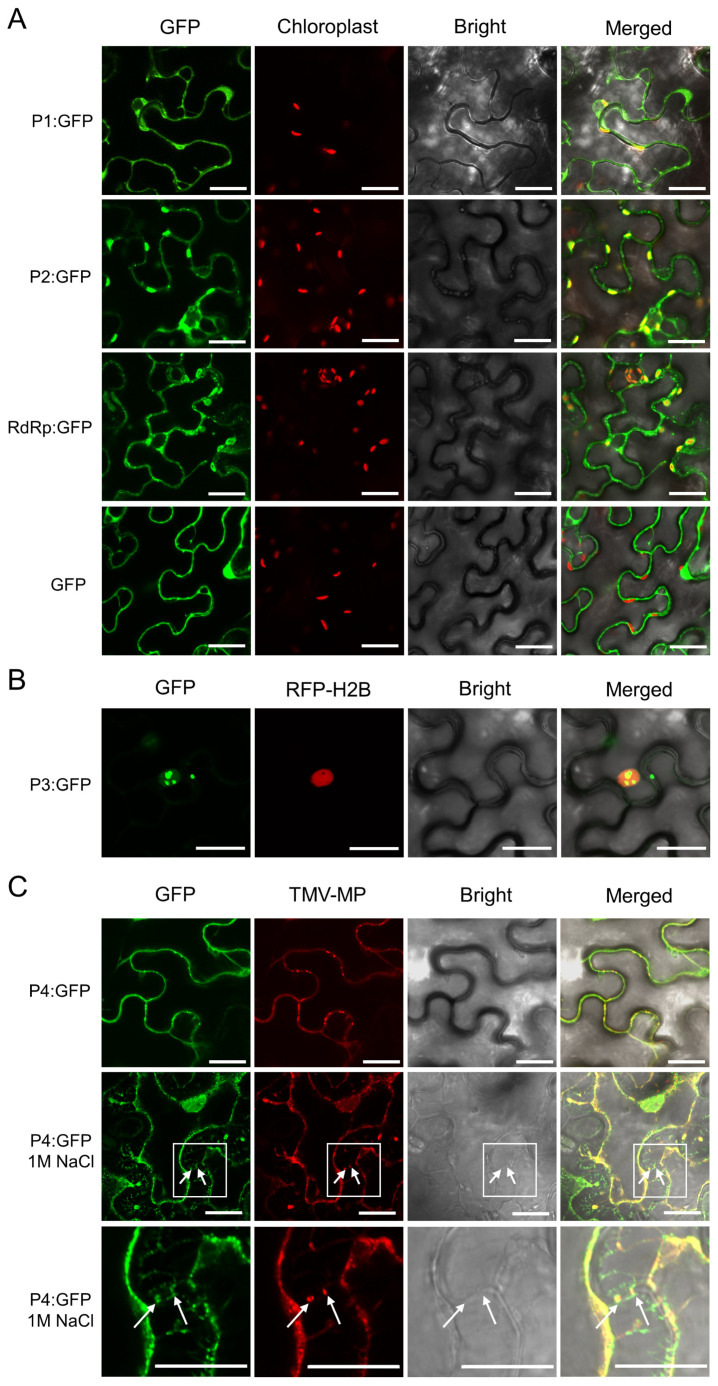
Subcellular localization of viral proteins of PSCYV was observed with confocal microscopy. (**A**) Subcellular localization of PSCYV P1, P2, and RdRp in infiltrated *N. benthamiana* leaves (green). Red signal represents chloroplast. (**B**) Subcellular localization of PSCYV P3 in infiltrated *N. benthamiana* leaves (green). Red signal represents nucleus. (**C**) Colocalization of PSCYV P4 and TMV MP-RFP in co-infiltrated *N. benthamiana* leaves. The infiltrated *N. benthamiana* leaves were treated with 1 M NaCl at 2 dpi. Red signal represents the subcellular localization of TMV-MP. White arrows indicate PD position. The bottom row is the enlarged image of the boxed area in the middle image. Scale bars, 20 µm.

**Figure 5 viruses-14-01821-f005:**
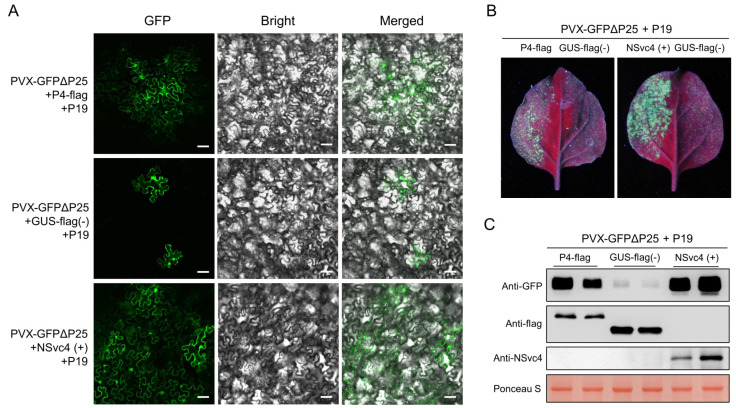
PSCYV P4 can complement the intercellular movement of mutant PVX-GFPΔP25. (**A**) PSCYV P4 can complement the intercellular movement of mutant PVX-GFPΔP25. The co-infiltrated *N. benthamiana* leaves were observed with a confocal microscope at 7 dpi. Bars, 50 μm. (**B**) GFP fluorescence of co-infiltrated *N. benthamiana* leaves was observed under a UV lamp at 7 dpi. (**C**) Western blot assays analyzing GFP and viral protein accumulation levels in co-infiltrated leaves. The large RuBisCO subunit stained by Ponceau S served as the loading control.

**Figure 6 viruses-14-01821-f006:**
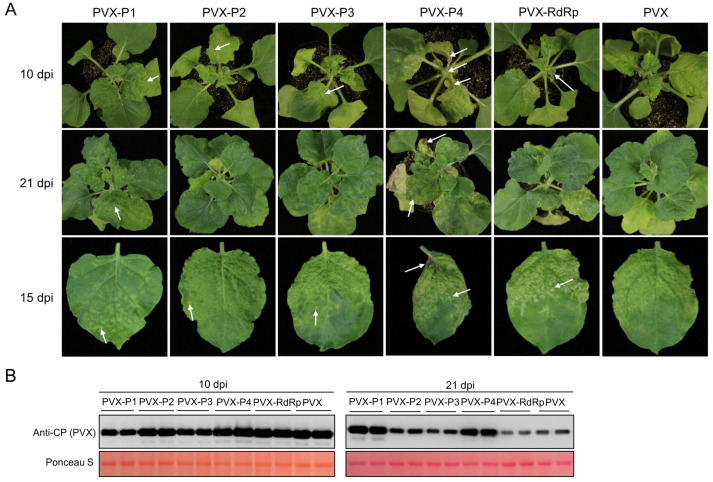
Symptom determinant analysis of PSCYV using the PVX heterologous expression system. (**A**) Symptoms of *N. benthamiana* plants expressing PSCYV P1, P2, P3, P4, or RdRp using the PVX expression vector. White arrows indicate necrotic sites. (**B**) Western blot assay analyses of PVX CP accumulation levels in infected *N. benthamiana* plant leaves at 10 dpi and 21 dpi. The large RuBisCO subunit stained by Ponceau S serves as the loading control.

**Figure 7 viruses-14-01821-f007:**
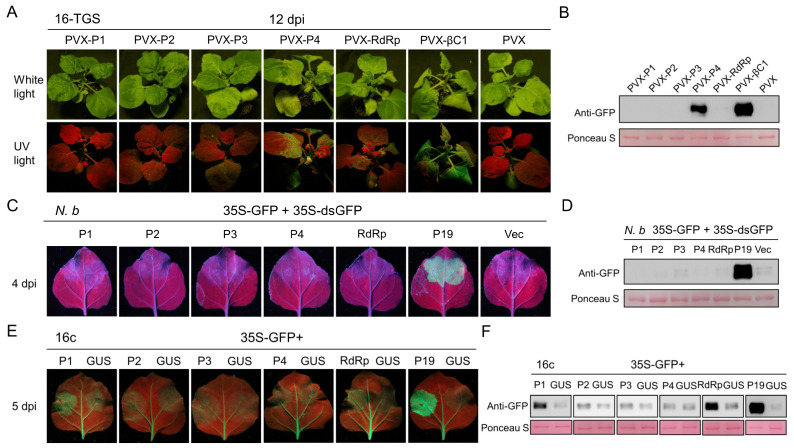
P1, RdRp, and P4 of PSCYV function as a suppressor of RNA silencing. (**A**) Photographs of 16c-TGS *N. benthamiana* plants inoculated with PVX-P1, PVX-P2, PVX-P3, PVX-P4, PVX-RdRp, PVX-βC1 (the positive control), or PVX (the negative control) under white light or UV light at 12 dpi. (**B**) Western blot assay analyzing GFP accumulation levels in the leaves from (**A**). (**C**) *N. benthamiana* leaves infiltrated with a mixture of *A. tumefaciens* expressing GFP (35S-GFP) and double-stranded GFP (35S-dsGFP), and P1, P2, P3, P4, RdRp, P19 (the positive control), or Vec (Vector, the negative control) were photographed under UV light at 4 dpi. (**D**) Western blot assay showing GFP accumulation levels in the leaves from (**C**). (**E**) The leaves of 16c *N. benthamiana* plants co-infiltrated with *A. tumefaciens* cultures expressing GFP (35S-GFP) and P1-flag, P2-flag, P3-flag, P4-flag, RdRp-flag, P19 (the positive control), or GUS-flag (the negative control) were photographed under UV light at 5 dpi. (**F**) Western blot assays analyzing GFP accumulation levels in the leaves from (**E**). The large RuBisCO subunit stained by Ponceau S serves as the loading control.

**Figure 8 viruses-14-01821-f008:**
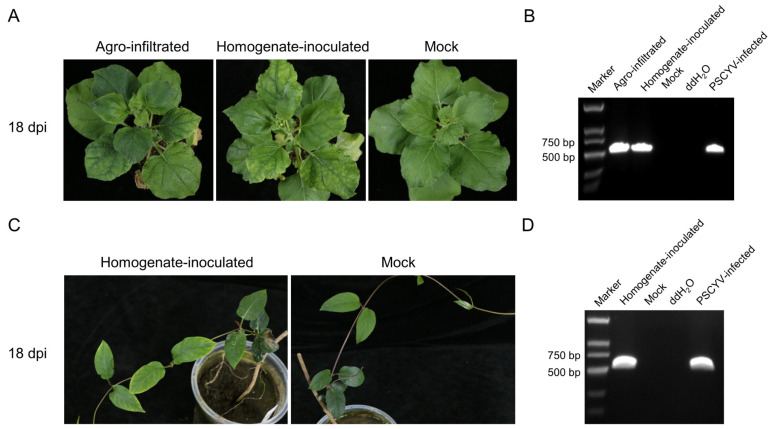
Agroinfiltration inoculation of the infectious clone and mechanical inoculation of PSCYV. (**A**) Symptoms of *N. benthamiana* plants inoculated with the infectious clone of PSCYV (**left**) and the homogenate of PSCYV-infected *Paederia scandens* (**middle**) compared with the mock inoculated plant (**right**). *N. benthamiana* plants were infiltrated with *A. tumefaciens* cultures carrying pCB301-PSCYV, or mechanically inoculated with the homogenate from the field-collected PSCYV-infected *Paederia scandens* plant. (**B**) RT-PCR detection of PSCYV in the plants as shown in (**A**). (**C**) Symptoms of mechanically inoculated *Paederia scandens* plants with the homogenate from agro-infiltrated *N. benthamiana* plant tissues. (**D**) RT-PCR detection of PSCYV in the plants shown in (**C**).

**Table 1 viruses-14-01821-t001:** Genomic RNA and viral protein sequence similarities between PSCYV and other ten members in the genus *Umbravirus*.

Virus Name	Nucleotide Identities (%)	Amino Acids Identities (%)				
	Genome	P1	P2	RdRp	P3	P4
Carrot mottle mimic virus	48.5	25.1	60.5	48.9	20.0	36.3
Carrot mottle virus	47.9	27.9	59.6	51.3	18.8	41.0
Ethiopian tobacco bushy top virus	55.0	31.7	71.6	57.3	36.6	56.6
Groundnut rosette virus	55.2	33.8	69.7	56.9	40.0	57.0
Ixeridium yellow mottle virus 2	49.5	29.0	59.8	50.3	27.9	39.8
Opium poppy mosaic virus	53.9	35.8	67.9	58.2	32.8	54.2
Patrinia mild mottle virus	49.6	26.1	59.4	50.7	24.8	39.3
Pea enation mosaic virus 2	50.1	25.8	61.2	50.1	31.0	57.6
Tobacco bushy top virus	55.6	31.4	71.2	59.2	32.8	58.4
Tobacco mottle virus	57.4	None	58.4	60.6	31.0	52.2

Red numbers indicate the highest similarity. None, no submitted sequence.

## Data Availability

Not applicable.

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
