# Peer review of "Discovery and Characterization of a Novel Umbravirus from Paederia scandens Plants Showing Leaf Chlorosis and Yellowing Symptoms"

_viruses, 2022, doi:10.3390/v14081821_

Round 1
Reviewer 1 Report
The manuscript describes the discovery of a novel virus from Paederia scandens through RNA sequencing and its molecular and functional characterization. Analysis of the genome and phylogenetic relationship grouped it as a new umbravirus which is tentatively named as Paederia scandens chlorosis yellow virus (PSCYV). Authors were able to construct a full cDNA clone of PSCYV from the affected plant which was infectious in Nicotiana benthamiana and were able to sap transmit PSCYV from N. benthamiana to Paederia scandens and proved Koch's postulates. Further, functional characterization of the encoded proteins involved in the virus movement, symptom development and RNA silencing were attempted and virus proteins, if any, with regard to the processes were identified. In addition to their findings on PSCYV this paper brings out some new information on the basic molecular biology of the umbraviruses in general. The manuscript is clearly stated and well written and I accept the article for publication in Viruses. Two minor edits noted below could be corrected at proof reading stage.
L 85: Based on
L223: Fig. 1 Legend- chlorosis
Author Response
The manuscript describes the discovery of a novel virus from Paederia scandens through RNA sequencing and its molecular and functional characterization. Analysis of the genome and phylogenetic relationship grouped it as a new umbravirus which is tentatively named as Paederia scandens chlorosis yellow virus (PSCYV). Authors were able to construct a full cDNA clone of PSCYV from the affected plant which was infectious in Nicotiana benthamiana and were able to sap transmit PSCYV from N. benthamiana to Paederia scandens and proved Koch's postulates. Further, functional characterization of the encoded proteins involved in the virus movement, symptom development and RNA silencing were attempted and virus proteins, if any, with regard to the processes were identified. In addition to their findings on PSCYV this paper brings out some new information on the basic molecular biology of the umbraviruses in general. The manuscript is clearly stated and well written and I accept the article for publication in Viruses. Two minor edits noted below could be corrected at proof reading stage.
Our response: We would like to thank the respected reviewer 1 for your positive comments and valuable suggestions.
- L 85: Based on
Our response: Revised accordingly.
- L223: Fig. 1 Legend- chlorosis
Our response: Revised accordingly.
Reviewer 2 Report
Manuscript viruses-1851844 describes the identification and characterization of Paederia scandens chlorosis yellow virus (PSCYV) in Paederia scandens plants in China. The research is of interest and is thoroughly conducted for the most part. Additionally, the manuscript is overall well written although it should be improved to address language issues. See some recommendation below. What is not acceptable is that no NCBI accession number of the PSCYV is made available to the reviewers although it is claimed that one will be obtained in two days (lines 213-214). Based on the fact that no NCBI accession number is provided with the submission, I recommend rejection of manuscript viruses-1851844 in its present form.
Specific comments.
Line 3: eliminate the
Line 21: Change the to a
Line 21 and throughout the manuscript: Change RdRp to RdRP
Line 24: ... (TGS) suppressor. Both P1 and ....
Line 28: Change the to a
Line 41: Eliminate other
Line 45: Eliminate their
Line 46: Eliminate the
Line 50-51: ... motifs of a viral ...
Line 82: ... leaf chlorosis and yellowing. In this study, we identified and characterize a novel virus from symptomatic Paederia scandens plants.
Line 82-94: Eliminate these sentences.
Line 13): Change organization to Oragnization
Line 139: ... that of known ...
Line 147-148: ... utilizing MUSCLE
Line 152: The predicted viral genes were amplified ...
Line 163: ... harboring two different constructs ...
Line 169: Change an to the
Line 186: ... homogenate for inoculation, diseases leaves ...
Line 186: Disease leaves from which plant were used as inoculum? Please clarify.
Line 187: Eliminate the
Line 201: Change sequencing to sequence
Line 202: ... a long contig of 293 ...
Author Response
Manuscript viruses-1851844 describes the identification and characterization of Paederia scandens chlorosis yellow virus (PSCYV) in Paederia scandens plants in China. The research is of interest and is thoroughly conducted for the most part. Additionally, the manuscript is overall well written although it should be improved to address language issues. See some recommendation below. What is not acceptable is that no NCBI accession number of the PSCYV is made available to the reviewers although it is claimed that one will be obtained in two days (lines 213-214). Based on the fact that no NCBI accession number is provided with the submission, I recommend rejection of manuscript viruses-1851844 in its present form.
Our response: We would like to thank the respected reviewer 2 for your positive comments and useful comments. As we declared in MS, the NCBI accession number of the PSCYV genomic sequence (OP053684) was obtained from GenBank and added to this revision.
Specific comments.
- Line 3: eliminate the
Our response: Revised accordingly.
- Line 21: Change the to a
Our response: Revised accordingly.
- Line 21 and throughout the manuscript: Change RdRp to RdRP
Our response: As written in general publications, the abbreviation of RNA-dependent RNA polymerase is RdRp, not RdRP. Thus, we think that RdRp is correct.
- Line 24: ... (TGS) suppressor. Both P1 and ....
Our response: Revised accordingly.
- Line 28: Change the to a
Our response: Revised accordingly.
- Line 41: Eliminate other
Our response: Revised accordingly.
- Line 45: Eliminate their
Our response: Revised accordingly.
- Line 46: Eliminate the
Our response: Revised accordingly.
- Line 50-51: ... motifs of a viral ...
Our response: Revised accordingly.
- Line 82: ... leaf chlorosis and yellowing. In this study, we identified and characterize a novel virus from symptomatic Paederia scandens plants.
Our response: Revised accordingly.
- Line 82-94: Eliminate these sentences.
Our response: Revised accordingly.
- Line 131: Change organization to Oragnization
Our response: Revised accordingly.
- Line 139: ... that of known ...
Our response: Revised accordingly.
- Line 147-148: ... utilizing MUSCLE
Our response: Revised accordingly.
- Line 152: The predicted viral genes were amplified ...
Our response: Revised accordingly.
- Line 163: ... harboring two different constructs ...
Our response: Revised accordingly.
- Line 169: Change an to the
Our response: Revised accordingly.
- Line 186: ... homogenate for inoculation, diseases leaves ...
Our response: Revised accordingly.
- Line 186: Disease leaves from which plant were used as inoculum? Please clarify.
Our response: Diseased leaves from the field-collected PSCYV-infected Paederia scandens plant or the agro-infiltrated N. benthamiana plants with typical symptoms were used as inoculum. Lines 187-188, in this revision, we have added the content: from the field-collected PSCYV-infected Paederia scandens plant or the agro-infiltrated N. benthamiana plants.
- Line 187: Eliminate the
Our response: Revised accordingly.
- Line 201: Change sequencing to sequence
Our response: Revised accordingly.
- Line 202: ... a long contig of 293 ...
Our response: Revised accordingly.
Round 2
Reviewer 2 Report
Revisions were satisfactorily addressed for the most part.